# An Adolescent Boy with Klinefelter Syndrome and 47,XXY/46,XX Mosaicism: Case Report and Review of Literature

**DOI:** 10.3390/genes13050744

**Published:** 2022-04-23

**Authors:** Tinka Hovnik, Eva Zitnik, Magdalena Avbelj Stefanija, Sara Bertok, Katarina Sedej, Vesna Bancic Silva, Tadej Battelino, Urh Groselj

**Affiliations:** 1Clinical Institute for Special Laboratory Diagnostics, University Children’s Hospital, UMC, 1000 Ljubljana, Slovenia; tinka.hovnik@kclj.si; 2Institute of Biochemistry and Molecular Genetics, Faculty of Medicine, University of Ljubljana, 1000 Ljubljana, Slovenia; 3Department of Pediatric Endocrinology, Diabetes and Metabolic Diseases, University Children’s Hospital, UMC, 1000 Ljubljana, Slovenia; eva.zitnik@kclj.si (E.Z.); magdalena.avbelj@mf.uni-lj.si (M.A.S.); sara.bertok@kclj.si (S.B.); katarina.sedej@gmail.com (K.S.); vesna.bancic@gmail.com (V.B.S.); tadej.battelino@mf.uni-lj.si (T.B.); 4Department of Pediatrics, Faculty of Medicine, University of Ljubljana, 1000 Ljubljana, Slovenia

**Keywords:** Klinefelter syndrome, mosaicism, 47,XXY/46,XX, review

## Abstract

Klinefelter syndrome is the most commonly reported sex chromosome abnormality. It is heavily underdiagnosed due to the substantial variability of clinical presentations but is generally characterized by small, firm testes, hypergonadotropic hypogonadism, and the absence of spermatogenesis. Most patients with Klinefelter syndrome have a 47,XXY genotype. If they present with mosaicism, two different cell lines are usually identified, an aneuploid 47,XXY cell line and a normal male 46,XY cell line. There are very few cases of 47,XXY mosaicism with the additional female cell line 46,XX described in the literature. We report a case of an adolescent with the male phenotype and a rare variant mosaic 47,XXY/46,XX karyotype who presented with painless bilateral gynaecomastia. 47,XXY and 46,XX mosaic cell lines were identified with GTG-banding and further characterized using fluorescent in situ hybridization. We summarized the available clinical presentations of reported male patients with 47,XXY/46,XX mosaicism. To improve the clinical management and quality of life in individuals with rare and cryptic genomic imbalances, the genetic diagnosis would need to be extended to atypical cases.

## 1. Introduction

Klinefelter syndrome (KS) is the most commonly reported sex chromosome abnormality, with an incidence of 1 per 600 newborn males [1]. It was first identified in 1942 when Klinefelter et al. published a report describing nine male patients with a constellation of clinical features including small testes, gynecomastia, azoospermia, and elevated follicle-stimulating hormone levels [2]. In 1959, the previously described clinical presentation was linked to the presence of a supernumerary X chromosome [3].

KS affects the testicular function. The loss of germ cells, which begins in foetal life and accelerates through puberty, leads to fibrosis and hyalinization of seminiferous tubules and relative hyperplasia of Leydig cells. Those processes result in characteristic features of KS such as small, firm testes, hypergonadotropic hypogonadism, and the absence of spermatogenesis [4,5].

Despite the high prevalence, only 10% of cases are diagnosed before the age of 14, reaching approximately 25% in adulthood [6]. The condition is most likely underdiagnosed due to substantial clinical variability and also possible occult presentations [7]. Before puberty, they only present with subtle features such as long legs, cryptorchidism, learning difficulties, and delays in speech and motor development [5,6,8]. Sexual development may be normal, and when the initiation of puberty is appropriate, virilisation is usually sufficient [9]. The onset of puberty is generally timely, and the testes can initially grow, but they usually later decrease back to a prepubertal size [10]. During adolescence, 38–56% of patients develop painless bilateral gynaecomastia [5,11]. They may present with poor muscle bulk, sparse body hair, narrow shoulders, and broad hips. In adulthood, patients are most commonly affected by fertility problems and sexual dysfunction [12]. Due to hypergonadotropic hypogonadism, androgen (testosterone) replacement therapy is described as the typical treatment for KS males [13]. 

KS is a result of either maternal or, in approximately 50% of cases, paternal meiotic nondisjunction of the sex chromosomes during gametogenesis [5]. Nondisjunction of the sex chromosomes during the first few meiotic divisions of a developing 46,XY zygote or the loss of a sex chromosome due to anaphase lagging are thought to be the cause of mosaic forms of this syndrome [14]. While most KS cases (80%) are due to the congenital numerical chromosome aberration 47,XXY, approximately 20% have higher-grade chromosome aneuploidies (e.g., 48,XXXY, 49,XXXXY), additional Y chromosomes (e.g., 48,XXYY), and structurally abnormal sex chromosomes, or are mosaics for two or more different cell lines [5]. The vast majority of KS mosaics have two different cell lines, an aneuploid 47,XXY cell line and a normal male 46,XY cell line [15]. If the patient’s original karyotype was 46,XY, a later nondisjunctional event could give rise to a 47,XXY cell line. However, the origin of a 47,XXY cell line from a 46,XX cell line is not easily explainable. 

The phenotype of aneuploid and different mosaic cell lines vary from typical KS male to disorders of sex development (DSD) from true hermaphrodite, ovarian hypoplasia, ovotesticular DSD, or normal female [16]. In this case report, we present an adolescent male patient with Klinefelter syndrome and a rare mosaic 47,XXY/46,XX karyotype identified with GTG-banding and further characterized using fluorescent in situ hybridization (FISH). 

## 2. Case Report

The propositus, an adolescent with one-year history of gynecomastia, was referred to our clinic for evaluation. He was born as an only child, conceived on the second attempt of IVF to a 34-year-old mother. The reason for her infertility is unknown. The pregnancy was reported to be normal. He was born at term with a low birth weight of 2300 g. He presented with chordae, proximal penile hypospadias, and right-sided cryptorchidism, which were surgically repaired during the early childhood. At that time, the right testis and right spermatic cord were found to be hypoplastic, and the left testis appeared to be hypoplastic on palpation. In the first year of life, he had a febrile seizure. His psychomotor development was normal, as was his school performance. He had no known diseases or allergies.

On first examination at our clinic, he was 14 years old, his height was 176 cm (87p), and his weight was 52 kg (48p). Apart from gynecomastia and asthenic constitution, there was nothing specific about his aspect. Physical examination showed normal male genitalia after surgical correction; both testes were estimated to be 4 mL in size by Prader orchidometer and positioned in the scrotum. The pubertal stage after Tanner was P2-3, A2. Palpation of the chest revealed 4 × 4 × 2 cm bilateral non-tender gynecomastia. On later check-ups, the testes were estimated to be 3 mL in size, and the right testis was found to be firm in consistency and atrophied.

Our patient had breast tissue removed by a plastic surgeon. He was monitored by an endocrinologist, who started testosterone undecanoate depo therapy when his testosterone level dropped below a critical level (total testosterone 3.1 nmol/L (normal range 8.8–30.6)) at the age of 17.5 years. He was diagnosed with mild Hashimoto thyroiditis but remained euthyrotic during follow-up. His bone mineral density measured by dual energy X-ray absorptiometry (DXA) before the introduction of testosterone supplementation was within the normal range (whole body BMD Z score -0.5; Lumbar spine L1-L4 BMD Z-score 0.7). 

At the last check-up, he was 20 years old, his final height was 182.13 cm (76P), and his weight was low (BMI 18.3 kg/m^2^—5P). He was reaching normal testosterone values with 250 mg of testosterone undecanoate every 3 months.

## 3. Results

### 3.1. Laboratory Studies

Hormonal studies at 14 years old revealed primary hypogonadism: elevated follicle stimulating hormone FSH; 28.1 IU/L (0.3–3.5 IU/L) and luteinizing hormone LH; 23.4 IU/L (0.2–1.9 IU/L) levels, low total testosterone; 5.4 nmol/L (0.1–28.6 nmol/L) and free testosterone; 8.0 pmol/L (30.15–189.8 pmol/L) levels and oestradiol levels of 0.14 pmol/L. His serum β-hCG was in the normal range. Semen analysis was performed at 15 years of age, and no sperm were found.

### 3.2. Cytogenetic and Molecular Studies

Cytogenetic analysis of peripheral blood revealed the presence of two cell lines: a normal female 46,XX cell line in 91% of the examined cells and an aneuploid 47,XXY cell line in 9% of the examined cells. Karyotyping was performed by GTG banding with a resolution of 400–500 bp in a haploid set. Mosaicism was also confirmed with fluorescent in situ hybridisation (FISH) using probe LSI SRY/CEPX (Vysis, Abbott). Combined dual labelled probe hybridizes on the centromeric region of the X chromosome and on the specific SRY gene section on the Yp11.3 chromosome (Figure 1).

Among 121 metaphase cells from peripheral blood, 110 cells were 46,XX and 11 cells were 47,XXY, corresponding to 91% and 9% of cells, respectively. We determined the final genotype of our patient according to ISCN nomenclature 46,XX[29]/47,XXY[3].ish X(DXZ1x2,SRYx0)[81]/X(DXZ1x2),Y(SRYx1)[8]. PCR-based Y-microdeletion analysis successfully identified five Y-specific loci (SRY, AMGL, DYZ3, DYZ274, and DYZ1), confirming the presence of a Y chromosome. Karyotyping of the chromosomes from peripheral blood lymphocytes of the patient’s parents showed a normal female 46,XX in his mother and a normal male 46,XY karyotype in his father. Gonadal tissues were not available for the study.

## 4. Discussion

47,XXY Klinefelter syndrome is a result of meiotic nondisjunction of the sex chromosomes during gametogenesis. The origin of a 47,XXY cell line from a 46,XX cell line is not easily explainable. In the case of our patient, the most likely hypothesis, therefore, is that the 46,XX cell line was derived from a post-zygotic loss of a Y chromosome in a proportion of cells of a primordial 47,XXY zygote. In contrast, the loss of an X chromosome would give rise to a 46,XY cell line. The possibility of gonadal mosaicism in the parents, which could result in our patient’s genotype, is less probable.

There are 20 cases of 47,XXY/46,XX mosaicism with different phenotypical outcomes described in the literature. To our knowledge, eight cases of phenotypic male individuals with KS features [17,18,19,20,21,22,23,24] (Table 1), eight cases of ovotesticular disorder of DSD [25,26,27,28,29], one case of a phenotypic female with ovarian hypoplasia [30], one case of discordant sex in monozygotic twins [24], and one case of normal female phenotype [31] have been reported so far. A case of 47,XXY/46,XX mosaicism with cystic hygroma was diagnosed prenatally, but the pregnancy was electively terminated at 22 weeks [32]. 47,XXY/46,XX cell lines are also reported in patients with more than two different cell lines [18,33,34].

Previous research suggests that patients with higher-grade aneuploidies are more severely affected [35], whereas 47,XXY/46,XY mosaics commonly exhibit fewer clinical signs compared to non-mosaic KS patients. They have larger mean testicular volumes, milder endocrine abnormalities, and less frequently present with gynaecomastia and azoospermia [5,36].

Phenotypic variations in non-mosaic KS could be explained by several factors. First is the X-linked androgen receptor (AR), which is linked to a polymorphic chain of cytosine-adenine-guanine (CAG) triplet repeats. The number of CAG repeats is inversely related to the activity of this receptor [37]. An increased repeat length would correlate negatively with testosterone’s effect [14]. Parental imprinting of the extra X chromosome and variable inactivation of some X-chromosomal genes may also be associated with the variability of the phenotype [14].

In mosaic KS, the degree to which different tissues are affected by mosaicism could be the key factor of clinical variability in these patients [37]. Tachon et al. reported a case of monozygotic 47,XXY/46,XX twins with discordant sex. They performed a cytogenetic analysis of peripheral blood lymphocytes, buccal smears, and urinary sediments, which showed different proportions of 47,XXY and 46,XX cell lines in different tissues and also differed between twins. There was a much higher percentage of the 46,XX cell line present in the buccal smear and urinary sediment of the phenotypic girl [24].

The development of the testes is determined by the presence or absence of a Y chromosome. In mosaics, the differentiation of gonads to testes or ovaria depends on the predominating cell line in the gonads [17]. In our patient, who presented with male external genitalia, cytogenetic analysis of peripheral lymphocytes showed a 46,XX cell line in 91% of the examined cells. Karyotyping of gonadal cells would be necessary to reveal the prevailing cell line in the patient’s testes; however, this was not performed as no further diagnostic value for the patient was expected.

Klinefelter syndrome and other related anomalies are heavily underdiagnosed, likely due to their clinical variability and also due to possible occult and rare presentations. It affects the testicular function, which results in primary hypogonadism. Patients can present with a variety of symptoms due to insufficient testosterone levels: poor muscle bulk, sparse body hair, narrow shoulders and broad hips, underdevelopment or regression of secondary sexual characteristics, osteoporosis, depression, and diminished intellectual capacity. In adulthood, patients are most commonly affected by fertility problems and sexual dysfunction [12,38]. Another typical symptom is breast growth, as in our case. Such cases have a slightly higher risk of malignant transformation. They also have an increased risk of some autoimmune diseases, particularly those that are female-predominant [16].

The identification of spermatozoa in sperm is extremely rare in KS patients. While in adolescents, spermatozoa are extracted from sperm anecdotally, in only 0–4%, a slightly better outcome (9%) is obtained in young adults [39]. Nevertheless, the expectations to father children are not that pessimistic. By using the invasive procedure testicular sperm extraction by microdissection of seminiferous tubules (micro-TESE), viable spermatozoa are obtained in about 50% of adults with KS [40], which could be used to fertilize egg cells by intracytoplasmic sperm injection (ICSI). The micro-TESE outcomes are worse in younger adolescents, where mostly only spermatogonia, sperm cell precursors, are collected; however, in vitro maturation of spermatogonia into mature spermatozoa remains a challenge. Therefore, some authors advise against performing micro-TESE before 16 years of age [39], while others claim that the optimal age needs to be adapted to the medical context and psychological maturity of each young man [41]. It is not clear what the optimal timing is for micro-TESE and whether and when the existing spotty sperm production would deteriorate. Prognostic parameters for a higher success rate of sperm retrieval include a higher preoperative testosterone level, higher preoperative testosterone/estradiol ratio [42,43], and a younger age (below 32 years of age) [42,44]. In our patient, micro-TESE was considered and planed when fertility would be desired by the couple. Micro-TESE is more successful in mosaic KS as compared to the classical karyotype [45], but the data on mosaic patients is scarce.

Based on our experience, active searching for Klinefelter syndrome symptoms is simple and effective and therefore sensible. Simple screening by palpation and volume estimation of the testes as part of a standard health examination of (pre)pubertal boys is neither difficult nor expensive. We believe that examination of the testes should be an essential part of preventive health checks during childhood and adolescence.

In conclusion, there are very few cases of 47,XXY mosaicism with an additional female cell line 46,XX described in the literature. We report a case of an adolescent with male phenotype and a rare variant mosaic 47,XXY/46,XX karyotype who presented with painless bilateral gynaecomastia. 47,XXY and 46,XX mosaic cell lines were identified with GTG-banding and further characterized using fluorescent in situ hybridization. We summarized the available clinical presentations of reported male patients with 47,XXY/46,XX mosaicism. To improve the clinical management and quality of life in individuals with rare and cryptic genomic imbalances, the genetic diagnosis would need to be extended to atypical cases.

## Figures and Tables

**Figure 1 genes-13-00744-f001:**
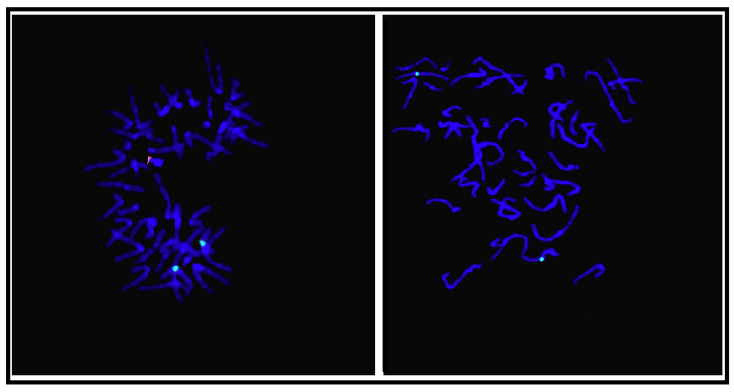
FISH analysis showing mosaic karyotype 47,XXY/46,XX using dual-colour FISH probe—Vysis LSI SRY (Spectrum Orange)/CEPX (Spectrum Green) (Abbott Molecular).

**Table 1 genes-13-00744-t001:** Clinical characteristics of published cases of phenotypic males with 47,XXY/46,XX mosaicism.

Published Reference	Age	Presented with	Testes Volume (mL or mm)	Height	Gynaecomastia	Malformations of GU Tract	Motor and Mental Development	FSH Levels
Song et al., 2014 [17]	18	Mediastinal germcell tumour	Small (no exact data)	No data	Mild	No data	No data	35.5 mIU/mL
Velissariou et al., 2006 [18]	29	Infertility	10 mm diameter	185 cm	Not present	None	Normal	12.9 mIU/mL
Mustaki et al., 1999 [19]	62	Teratoma of the right testis	Left testis: 1 mL	170 cm	Not present	None	No data, normal	69 mIU/mL
Mohd et al., 2016 [20]	12	Eunuchoid body habitus	Left testis: 2 mL	148 cm	Mild gynaecomastia	Right cryptorchidism	Normal	1.4 mIU/mL
Kanaka-Gantenbein et al., 2007 [21]	13	Left scrotal hemorrahage	Left: ovary Right: testis	169 cm	Significant gynecomastia	Normal ovary	Normal	20 mIU/mL
Isguven et al., 2005 [22]	14	Sweeling in hemiscrotum	Right: 3 mL, Left: 4–5 mL	170 cm	Bilateral gynecomastia	hypospadias	Normal	29.4 mUI/mL
Perez-Palacios et al, 1981 [23]	16	Unilateral cryptorchidism	Right: 25 × 10 × 10 Left: missing	144 cm	Bilateral gynecomastia	Small uterus	Normal	5.5 mIU/mL
Tachon et al., 2014 [24] *	5	Speech delay in male tween	Left: 14 × 9 × 6 Right: 14 × 10 × 6	110 cm	Not present	None at age of 5	Normal	1.8 mUI/mL

* All data at the age of five years old.

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
