# Peer review of "An Adolescent Boy with Klinefelter Syndrome and 47,XXY/46,XX Mosaicism: Case Report and Review of Literature"

_genes, 2022, doi:10.3390/genes13050744_

Round 1

Reviewer 1 Report

The manuscript An adolescent boy with Klinefelter syndrome and 47,XXY/46,XX mosaicism: case report and review of literature describes a rare mosaic geno- and phenotype of 47,XXY, Klinefelter syndrome. This manuscript contributes novel explanatory evaluation of the genotypic presentation versus phenotypic presentation of this rare genetic diagnosis.

Introduction

  • Authors state that males with 47,XXY present with “mild cognitive impairment.” This information is outdated, as much of the current research demonstrates intact (non-verbal) intelligence in 47,XXY. The manuscript should be modified to cite delays in speech, language, and motor development.
  • Authors may want to briefly mention hormonal replacement therapy as a typical treatment for the associated androgen deficiency in 47,XXY.

Case Report

  • It may be beneficial to the manuscript to briefly review demographics, socioeconomic status, family history, genetic tests, and any other possible affecting factors of the patient that may contribute to the propositus’s clinical presentation and/or mother’s infertility.

Discussion

  • This reviewer believes the conclusions drawn on this case report were well supported by the existing literature. Authors sufficiently and appropriately mentioned the study’s limitations, clinical implications, and future directions.

Overall, this manuscript will provide valuable insight to the rare 47,XXY/46,XX karyotype after authors address our above comments. We would be happy to re-review when the changes are made.

Author Response

The response to the Reviewer 1 is attached.

Reviewer 2 Report

Dear authors, congratulations for your manuscript. It is true, Klinefelter syndrome should not be that difficult to diagnose, nevertheless it is misdiagnosed in various situations, hence your work is reminding all of us of this entity and is putting us in high alert.

My main remarks upon your work have to do with the case presentation and to be specific i think that you should provide adequate information regarding the primary and following visits of your patient. You have mentioned that upon his first visit he was 14 tears old and his external genitalia were normal but on later check-ups you discovered an atrophied right testis and a reduced in size left one. So my questions can be summed-up in:

  1. What was the age of the patient at the last checkup?
  2. Have you ever considered  micro-TESE since azoospermia was revealed? If yes at what age?
  3. You report that an andrologist suggested cryopreservation of semen, but you mentioned earlier that the first semen analysis revealed azoospermia. If that was the case, didi you perform only one semen analysis at the primary estimation or more than one, and did you perform another after what period of time?
  4. Regarding cryopreservation many authors suggest that should not be proposed at under the age of 16 (Franik S, Hoeijmakers Y, D'Hauwers K, Braat DD, Nelen WL, Smeets D, Claahsen-van der Grinten HL, Ramos L, Fleischer K. Klinefelter syndrome and fertility: sperm preservation should not be offered to children with Klinefelter syndrome. Hum Reprod. 2016 Sep;31(9):1952-9. doi: 10.1093/humrep/dew179. Epub 2016 Jul 13. PMID: 27412247.) and others claim that the optimal age for proposing the first sperm collection could be adapted according to the medical context and the psychological maturity of the young man  (Ly A, Sermondade N, Brioude F, Berthaut I, Bachelot A, Hamid RH, Khattabi LE, Prades M, Lévy R, Dupont C. Fertility preservation in young men with Klinefelter syndrome: A systematic review. J Gynecol Obstet Hum Reprod. 2021 Nov;50(9):102177. doi: 10.1016/j.jogoh.2021.102177. Epub 2021 Jun 1. PMID: 34087451.)

As you see I am focusing on fertility issues, since pretty much you have covered all the other issues apart his sex life.

Should you provide more details and adress tha issue at he discussion session, I trust that you will provide a thorough analysis of the entity you are trying to present.

Author Response

The response to the Reviewer 2 is attached.
